# BALSA: integrated secondary analysis for whole-genome and whole-exome sequencing, accelerated by GPU

Ruibang Luo[1,3], Yiu-Lun Wong[1,3], Wai-Chun Law[1,3], Lap-Kei Lee[1,2], Jeanno Cheung[1], Chi-Man Liu[1] and Tak-Wah Lam[1]

[1] HKU-BGI Bioinformatics Algorithms and Core Technology Research Laboratory & Department of Computer Science, University of Hong Kong, Hong Kong
[2] School of Science and Technology, The Open University of Hong Kong, Hong Kong
[3] These authors contributed equally to this work.

## ABSTRACT

This paper reports an integrated solution, called BALSA, for the secondary analysis of next generation sequencing data; it exploits the computational power of GPU and an intricate memory management to give a fast and accurate analysis. From raw reads to variants (including SNPs and Indels), BALSA, using just a single computing node with a commodity GPU board, takes 5.5 h to process 50-fold whole genome sequencing (∼750 million 100 bp paired-end reads), or just 25 min for 210-fold whole exome sequencing. BALSA's speed is rooted at its parallel algorithms to effectively exploit a GPU to speed up processes like alignment, realignment and statistical testing. BALSA incorporates a 16-genotype model to support the calling of SNPs and Indels and achieves competitive variant calling accuracy and sensitivity when compared to the ensemble of six popular variant callers. BALSA also supports efficient identification of somatic SNVs and CNVs; experiments showed that BALSA recovers all the previously validated somatic SNVs and CNVs, and it is more sensitive for somatic Indel detection. BALSA outputs variants in VCF format. A pileup-like SNAPSHOT format, while maintaining the same fidelity as BAM in variant calling, enables efficient storage and indexing, and facilitates the App development of downstream analyses. BALSA is available at: http://sourceforge.net/p/balsa.

Corresponding author
Tak-Wah Lam, twlam@cs.hku.hk

Subjects Bioinformatics, Genomics, Computational Science
Keywords Secondary analysis, Whole-genome seqeuncing, Whole-exome sequencing, GPU, Variant calling, Genomics, NGS, HPC

## INTRODUCTION

With the advance in next generation sequencing (NGS) technologies, whole exome sequencing (WES) and whole genome sequencing (WGS) have become compelling tools for clinical diagnosis and genetic risk prediction. Sequencing data requires dedicated analysis tools to produce a robust characterization before being used by scientists or clinicians. To this end, analysis pipelines such as Baylor's Mercury (*Reid et al., 2014*) and those commercially available in DNAnexus and Seven Bridges Genomics have been developed. These pipelines take an automated approach to integrate multiple well-known open-source analysis components. Cost aside, Mercury reported finishing the analyses

of a WES human sample in 15 h using one computer node, and a WGS human sample (NA12878) in approximately 32 h using 8 computing nodes at peak. These pipelines have been deployed on public cloud services such as Amazon Web Services (AWS), which provides the hardware elasticity to analyze up to tens of thousands of samples simultaneously.

The cost and speed of NGS have been improving much faster than those of computer hardware. As recently announced by Illumina, sequencing cost is approaching the so-called "mythical" rate of $1,000 per whole genome sequencing. Many laboratories and hospitals nowadays routinely generate terabytes of NGS data daily; apart from sequencing, computational resources for running the above-mentioned analysis pipelines are indeed a major expenditure. The running cost, theoretically speaking, increases linearly with the running time and number of computing nodes required. Yet, in practice, the long running time of such pipelines often coupled with a lot of extra cost to fix possible errors due to nodes failure or corruption of intermediate data between the component tools, and to solve unexpected compatibility issues among component tools. Thus a single tool that is well designed to embrace the functionalities of all necessary components involved in NGS secondary analysis while being efficient even on a single node is promptly needed. The tool shall take raw reads as input, and outputs variants with sensitivity and accuracy competitive to or better than the prevalently utilized combinations of short-read aligners and variant callers. The tool shall have the extra capability to output the details of every single genome position in a space-efficient manner to facilitate users from recurring the analysis and to co-analyze with copious amount of samples. From the efficiency perspective, the tool shall be meticulously designed to maximize the utilization of every subsystem of a computing node.

Our previous work on short-read alignment, SOAP3-dp (*Luo et al., 2013*), which fits the problem of aligning individual reads with the massive parallelism provided by a Graphics Processing Unit (GPU), successfully solves the problem by two to tens of times faster than state-of-the-art short-read aligners, while maintaining the highest sensitivity and accuracy with read length of 100 bp and 150 bp. However, the acceleration is inadequate for the whole secondary analysis. For a typical WGS sample, SOAP3-dp can shorten the alignment time to 2–4 h, yet the follow-up analyses, which include base-score recalibration, de-duplication, realignment and variant calling procedures, still require tens of hours using a single computing node.

We have developed BALSA, a lightweight total solution for NGS secondary analysis that takes full advantage of the computational power available on a computing node equipped with a multi-core CPU and a GPU device. We have tested BALSA on a node equipped with a 6-core Intel i7-3930k, 64 GB 1333 MHz memory and an Nvidia GTX680 GPU with 4 GB memory, the end-to-end time to process a 50-fold WGS human dataset (~150 Gigabases) from FASTQ files into a VCF file of recalibrated variants with a "SNAPSHOT" of details per genome position is 5.5 h and can be as fast as 3 h on newer and professional models of CPU and GPU. A 210-fold WES human dataset takes 24.65 min with the same setting.

PeerJ ___________________

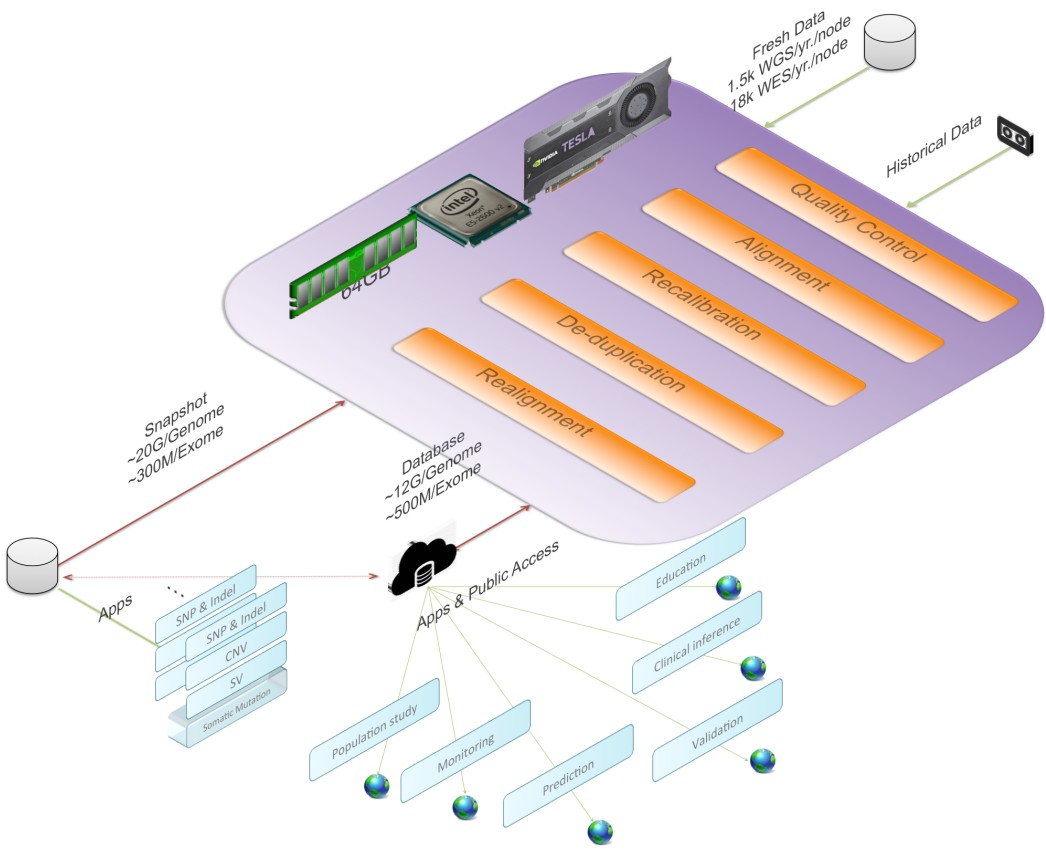

**Figure 1** **BALSA, based on SOAP3-dp, performs the whole secondary analysis (raw reads to variants) in memory with most of the modules accelerated with GPU.**

BALSA outperforms existing pipelines when considering the sensitivity and accuracy of detecting known variants in simulated data. It generates less SNP conflicts for a deeply sequenced trio family. BALSA's performance stems from using the 16-genotype model that incorporates both SNPs and Indels simultaneously in a diploid space, and its proactive and exhaustive realignment that maximizes the local variant signal coherency.

Figure 1 gives an overview of BALSA and Fig. 2 gives a flowchart of the pipeline of BALSA. BALSA extends our aligner SOAP3-dp so that while the GPU is aligning the reads, the CPU is processing the alignment results in the memory in parallel. Furthermore, BALSA is able to utilize the GPU for different computational intensive work, such as the exhaustive realignment of reads due to different hypothetical Indels in the reference genome. The speed advantage of BALSA is not entirely due to the GPU; BALSA has intricate memory management to minimize the use of hard disk. In a typical WGS sample, the reads and their alignment results would occupy hundreds of Gigabytes or even Terabytes. BALSA, with a succinct representation of the alignment results, is able to process all the reads for the purpose of variant calling almost entirely in the main memory. Processes like the removal of duplicate reads can be done without sorting a large volume of data records on the hard disk. A SNAPSHOT that is considered as a pileup

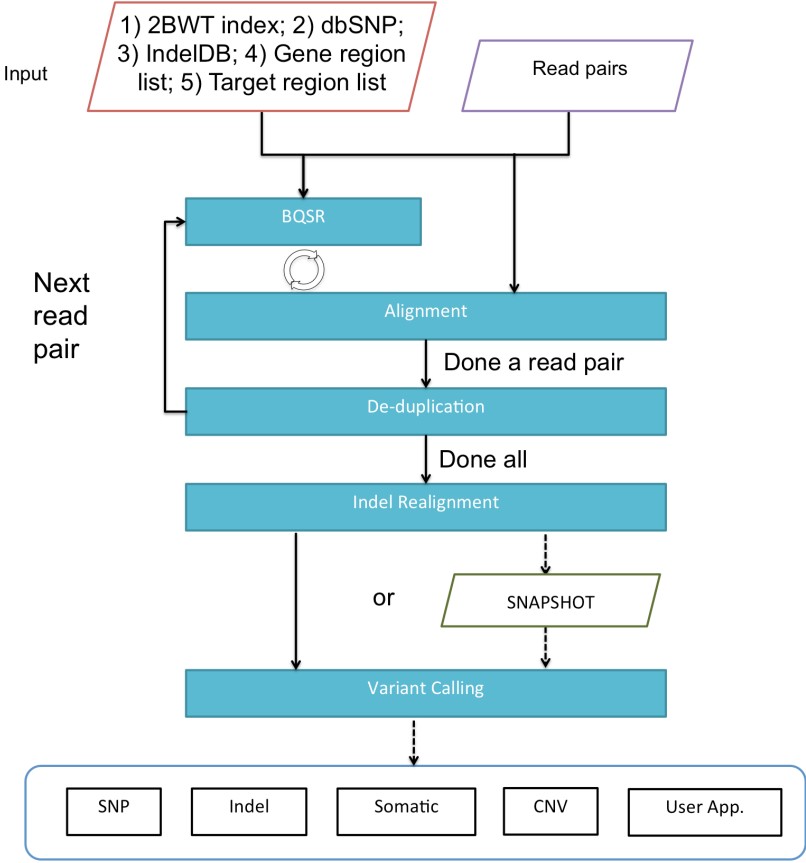

**Figure 2 A flowchart of the pipeline of BALSA.** BQSR denotes "base quality score recalibration".

form of all the information from the inputting raw reads after alignment, recalibration and realignment, is a simple dump of all the data structures that BALSA needs to work with, which enables repeatable variant calling without reprocessing the reads. The details of BALSA's algorithms and implementations are given in the Supplemental Information. BALSA has been optimized for Illumina platform, but the workflow can be adapted to other platforms such as Ion Proton.

## RESULTS

To demonstrate the performance of BALSA, we compare its speed and the quality of the identified variants to other pipelines (Table 1), which typically comprise (1) an aligner, (2) post-processing tools, and (3) a variant caller. We also compare BALSA to a recently published CPU-based integrated workflow named ISAAC (*Raczy et al., 2013*).

### Speed for WGS–YH 50-fold 100 bp paired-end reads

First of all, we compare the speed of BALSA, BWA + GATK (*DePristo et al., 2011*), SOAP3-dp + GATK, and ISAAC on real data. In particular, 50-fold 100 bp paired-end reads of the YH sample (*Luo et al., 2012*) (EBI SRA Accession: ERP001652, Appendix 1.1) were used (see Appendix 2 for the settings and commands). For BWA v0.7.5a, both the 'aln'

**Table 1** Aligners, post-processing tools, variant callers and integrated pipelines used for comparison with BALSA.

| Step | Tool | Citation |
|---|---|---|
| Aligner | BWA-bwaaln | *Li et al. (2009)* |
| | BWA-bwamem | *Li (2013)* |
| | SOAP3-dp | *Luo et al. (2013)* |
| Post-processing | GATK | *DePristo et al. (2011)* |
| | Picard | http://picard.sourceforge.net |
| Variant caller | Atlas2 | *Challis et al. (2012)* |
| | Freebayes | *Garrison & Marth (2012)* |
| | GATK HaplotypeCaller | *DePristo et al. (2011)* |
| | GATK UnifiedGenotyper | *DePristo et al. (2011)* |
| | Samtools | *Li et al. (2009)* |
| | Mutect | *Cibulskis et al. (2013)* |
| | Varscan | *Koboldt et al. (2012)* |
| Pipeline | ISAAC | *Raczy et al. (2013)* |
| Somatic Caller | Mutect | *Cibulskis et al. (2013)* |
| | SomaticSniper | *Larson et al. (2012)* |

version (*Li & Durbin, 2009*) and the new 'mem' version (*Li, 2013*) (with improved speed and sensitivity) were tested, and for GATK, we used best practice v4. The variant caller used is GATK UnifiedGenotyper. All experiments were performed on a computing node with a 6-core CPU (Intel i7-3930k@3.2GHz), 64 GB memory, and an Nvidia GTX680 GPU. The time reported is the average time over two repeated runs of each experiment.

In summary, from raw reads to variants (including SNPs and Indels), BALSA finished in 5.49 h, whereas ISAAC finished in 11.92 h, and GATK coupled with BWAaln, BWAmem and SOAP3-dp in 88.00, 48.68 and 46.27 h, respectively. See Fig. 3 for a comparison, and Table 2 for a breakdown of the running time. Although the overall time used by BWAmem + GATK and SOAP3-dp + GATK is similar, the alignment time of SOAP3-dp is indeed much shorter than BWAmem (4.12 h versus 14.56 h). BWAaln is the longest (46.16 h). SOAP3-dp's ability to identify more Indel candidatures causes GATK to run 8 more hours.

### Alignment & variant calling statistics

BALSA (and SOAP3-dp) has the highest alignment sensitivity. When measuring the number of read pairs that have both ends aligned and paired, SOAP3-dp/BALSA reports 97.08%, BWAmem 95.74%, BWAaln 92.22% and ISAAC 91.42% (see Table S1 for details). Table 3 shows the statistics of the variants (SNPS and Indels) called by different pipelines. For BALSA and GATK, we are able to count the raw SNPs called, as well as those SNPs that pass the VQSR filter with good variant quality, and those passed the filter but with low variant quality. BALSA reports a slightly higher number than GATK in each category (no matter GATK is coupled with BWAaln, BWAmem or SOAP3-dp). The Ti/Tv ratio, Ref Hets, and percentage of overlap with dbSNP are within normal ranges in all cases. The

**Table 2 Time consumption of different pipelines.** All number in hours.

| Step | BWAaln GATK + Picard UnifiedGenotyper | BWAmem GATK + Picard UnifiedGenotyper | SOAP3-dp GATK + Picard UnifiedGenotyper | ISAAC | BALSA |
|---|---|---|---|---|---|
| Alignment | 46.16 | 14.56 | 4.12 | | |
| Sort and merge | 1.40 | 1.70 | 1.74 | | |
| Mark duplicate | 6.84 | 6.25 | 5.50 | | |
| Realigner target creator | 0.93 | 0.77 | 1.06 | 9.89 | 5.24 |
| Indel realigner | 10.89 | 7.37 | 15.70 | | |
| Base score recalibration | 5.20 | 4.75 | 4.91 | | |
| PrintReads | 12.17 | 9.92 | 9.47 | | |
| Variant calling | 4.41 | 3.37 | 3.77 | 2.03 | 0.24 |
| Total | 88.00 | 48.68 | 46.27 | 11.92 | 5.49 |

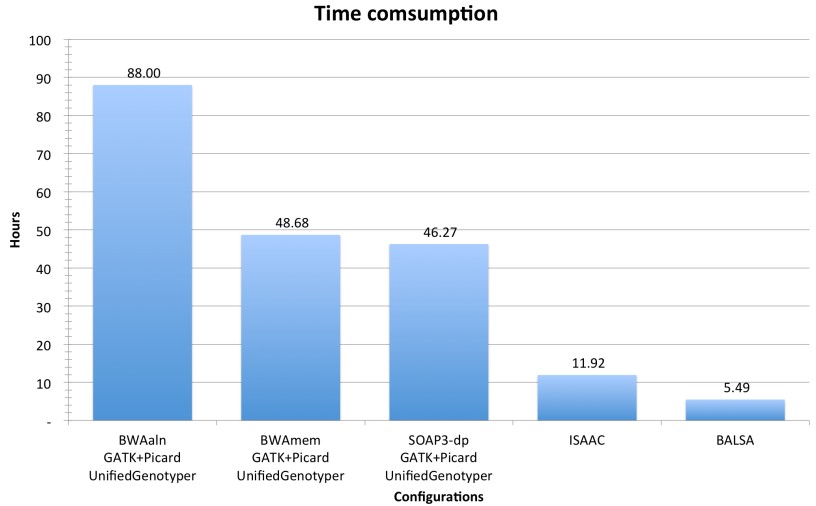

**Figure 3 Time consumption comparison between pipelines analyzing YH 50-fold 100 bp paired-end WGS data.**

Indel calling statistics is relatively more interesting. When counting Indels that can pass the VQSR filter (with good variant quality), BALSA detected 16.5%, 9.2% and 7.6% more than GATK coupled with BWAaln, BWAmem and SOAP3-dp, respectively. The increase over ISAAC is even more drastic. Note that the statistics reported here do not conclude the accuracy. In the next section we will use simulated data to study the accuracy and sensitivity of variant calling.

It is worth mentioning that BALSA comes with a Random Forest based filtration that can be used to replace the VQSR filtration (method and features used for model training described in Supplemental Information 8.3). The former costs only ∼15 min for a 50-fold WGS, while giving similar filtration power (in our experiment, 98.5% of the variants that pass the new filter (with probability ≥0.95) are overlapping with those variants passing the

**Table 3 Statistics of variants called by different pipelines.** "VQSR LowQual" means variants passed GATK VQSR but with (i) QUAL<50 for pipelines using UnifiedGenotyper and (ii) QUAL<30 for BALSA. "RandomForest LowQual" means variants with probability $\geq 0.95$ using random forest classification but with QUAL<30 for BALSA. Please refer to Supplemental Information 8.4.1 for the details of the variant QUAL profile of BALSA.

| Variant type | Metric | BWAaln GATK + Picard UnifiedGenotyper | BWAmem GATK + Picard UnifiedGenotyper | SOAP3-dp GATK + Picard UnifiedGenotyper | ISAAC | BALSA |
|---|---|---|---|---|---|---|
| SNP | Raw | 4,175,654 | 4,267,377 | 4,978,914 | 3,429,162 | 5,239,864 |
| | VQSR PASS | 3,324,891 | 3,307,619 | 3,383,853 | – | 3,444,915 |
| | VQSR LowQual | 151,933 | 136,392 | 308,321 | – | 877,964 |
| | RandomForest PASS | – | – | – | – | 3,433,397 |
| | RandomForest LowQual | – | – | – | – | 871,422 |
| | Ti/Tv | 2.08 | 2.07 | 2.05 | 2.08 | 2.04 |
| | dbSNP v137 | 99.62% | 99.47% | 98.60% | 99.29% | 98.51% |
| | Ref Hets | 54.40% | 54.40% | 55.40% | 57.20% | 58.20% |
| Indel | Raw (Indel) | 605,966 | 615,351 | 685,541 | 455,103 | 974,033 |
| | VQSR PASS | 576,889 | 615,351 | 624,629 | – | 671,914 |
| | RandomForest PASS | – | – | – | – | 630,827 |
| | dbSNP v137 | 90.70% | 90.49% | 87.80% | 93.38% | 89.01% |

VQSR filter. Figure 4 shows the correlation between the variant classification probabilities generated by BALSA and the VQSLOD value generated by GATK's VQSR.

## Sensitivity and accuracy for WGS—simulated data

To assess the accuracy and sensitivity of BALSA on variant calling, we used pIRS (*Hu et al., 2012*) short-read simulator to obtain a set of 40-fold Illumina-style 100 bp paired-end reads with 500 bp insert size, from a modified GRCh37 human reference genome with 2,859,141 known SNPs and 287,733 known Indels (Settings and commands elaborated in Appendix 1.2). We tested three different pipelines to process the simulated reads for variant calling: (1) BALSA, (2) SOAP3-dp + GATK + "6 prevalently used variant callers"[1] and (3) ISAAC. The results of the six variant callers were then combined to improve the sensitivity and accuracy of individual callers (see the rules in Appendix 2.4) to form an Ensemble call set, referred to as Ensemble below. Using one computing node (same configuration as above), BALSA and ISAAC finished in 3.86 and 8.71 h, respectively, whereas the Ensemble pipeline used more than a week (the time was dominated by the individual callers, which used about 5 days).

To make a fair comparison, no filtration was applied to the variants called by the three pipelines. Figure 5 compares the SNPs and Indels called by BALSA and Ensemble with respect to the correct SNPs and Indels covered by the simulated reads (denoted Truth below). Perhaps not surprisingly, Ensemble made more incorrect calls for SNPs and Indels and has higher False Discovery Rate (FDR) than BALSA (SNP: 0.21% versus 0.11%; Indel: 1.04% versus 0.34%), while Ensemble achieves higher sensitivity than BALSA, precisely,

[1] The variant callers tested include Atlas (*Challis et al., 2012*), Freebayes (*Garrison & Marth, 2012*), GATK HaplotypeCaller, GATL UnifiedGenotyper, Samtools (*Li et al., 2009*), and Mutect (only SNP) (*Cibulskis et al., 2013*)/Varscan (only Indel) (*Koboldt et al., 2012*). See Table 1. Note that in view of the results on real data, we have not tested BWA-based pipelines.

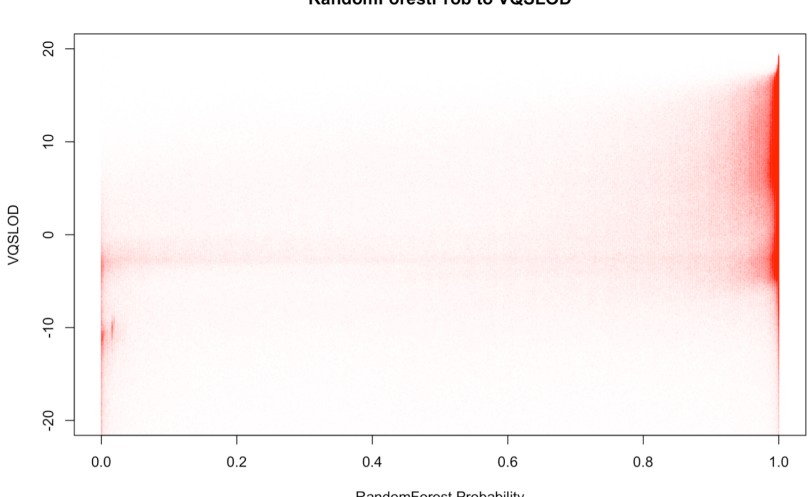

**Figure 4  Correlation plot between the RandomForest Probability generated by BALSA and the VQS-LOD value generated by GATK's VQSR on YH 50-fold 100 bp paired-end WGS data.**

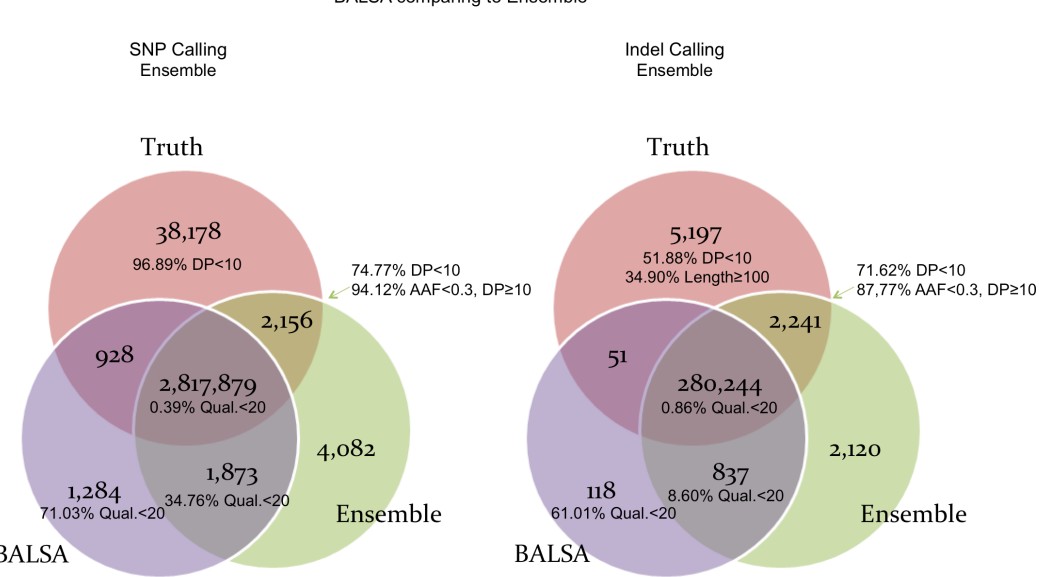

**Figure 5  Venn graphs illustrating the overlaps between (1) BALSA, (2) the Ensemble call set, and (3) the known variants on both SNP and Indel.** AAF denotes "alternative allele frequency", i.e., percentage of reads supporting the alternative allele among all simulated reads covering a variant. DP represents the number reads simulated covering a variant. Qual means the variant score assigned by BALSA.

0.04% and 0.76% higher in SNPs and Indels, respectively. Further investigation into the variants exclusively detected by Ensemble (2,156 SNPs and 2,241 Indels) indicated that 74.77% and 71.62% of such SNPs and Indels are covered with less than 10 reads generated from the simulation; and for the remainders supported by ≥10 reads, 94.12% and 88.77%

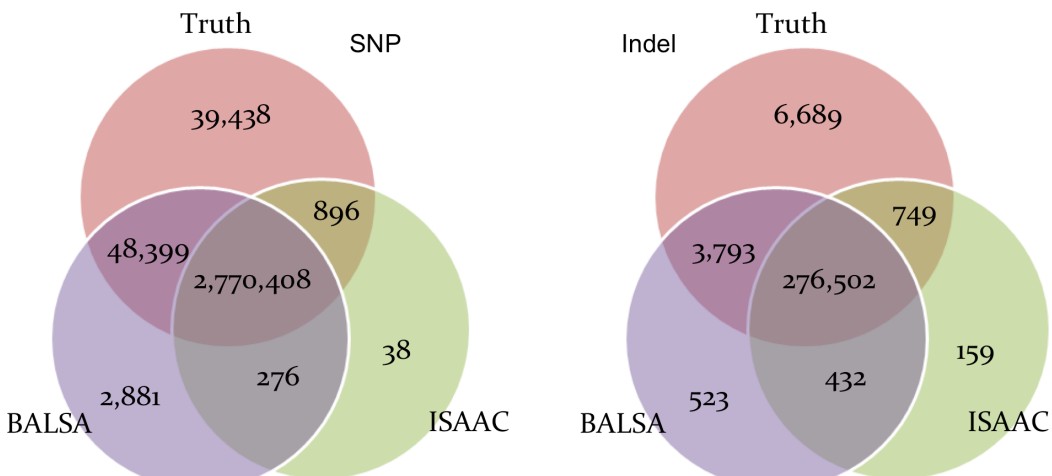

**Figure 6** Venn graphs illustrating the overlaps between (1) BALSA, (2) ISAAC, and (3) the known variants on both SNP and Indel.

of the SNPs and Indels are with alternative allele frequency (AAF) lower than 0.3. Hence we conclude that over 95% of the variants that are exclusively detected by Ensemble are unreliable and would eventually be filtered. Therefore, BALSA's sensitivity and accuracy is competitive to the combination of 6 prevalently used variant callers.

Figure 6 shows the comparison between BALSA and ISAAC. BALSA clearly outperformed ISAAC in terms of sensitivity (1.66% or 47,413 more SNPs, and 1.06% or 3,044 more Indels), and so did Ensemble. ISAAC's performance is probably limited by its alignment algorithm as its sensitivity is lower than all the other callers tested, where 3.67% less reads were aligned and 5.66% less reads were properly paired when compared to BALSA/SOAP3-dp. Notably, ISAAC has a slightly lower FDR than BALSA (0.10% and 0.12% lower for SNP and Indel, respectively).

In Figs. S1 and S2, BALSA is further compared with each of the six individual caller used in Ensemble. BALSA, while outperformed the 6 individual callers in either sensitivity or accuracy, achieved the best trade-off (SNP: Fig. S1, Indel: Fig. S2). We also compared the size distribution between simulated Indels and Indels detected by BALSA (Fig. S3).

## WGS–Trio study

To further test the accuracy of BALSA, SNP trio conflict analysis was performed. We used a trio from CEPH pedigree 1493, which consists of family members NA12877 (father, ERR091567-70, 54.59x), NA12878 (mother, ERR091571-74, 56.94x) and NA12882 (child, ERR091575-78, 54.18x), with data from Illumina Platinum Genome Project (*Illumina, 2012*). We measured the number of Mendelian SNP conflicts, each of which is a variant called in the child that is inconsistent with the genotypes of the parents. We run BALSA and BWAmem + GATK + UnifiedGenotyper on the three samples, with RandomForest

**Table 4** Run time, number of SNPs passing filter (with PASS tag), union of SNP sites and total number of SNP conflicts of BALSA and BWA + GATK for the NA12877, NA12878 and NA12882 family. Union is the SNP sites called in all samples or called in any sample.

| Pipeline | Sample | Time (Hour) | SNPs (PASS) | Union | Conflicts | Conflict rate |
|---|---|---|---|---|---|---|
| BALSA | NA12877 | 6.98 | 3,522,647 | | | |
| | NA12878 | 6.24 | 3,439,917 | 4,556,818 | 209,552 | 4.60% |
| | NA12882 | 6.65 | 3,428,070 | | | |
| BWA + GATK | NA12877 | 87.81 | 3,125,185 | | | |
| | NA12878 | 91.42 | 3,158,382 | 4,327,046 | 236,608 | 5.47% |
| | NA12882 | 87.01 | 3,183,451 | | | |

and VQSR filtration applied to the resulting variants, respectively (settings and commands elaborated in Appendix 2.1). The results of the two pipelines were transformed to gVCF format and analyzed by the trio conflict evaluation tool, which available as a part of the gvcftools (*Illumina, 2013*) package. BALSA took less than 20 h to analyze the three samples by using just a single computing node (same configuration as above), while the GATK pipeline use 266 h.

Results are shown in Table 4. As expected, BALSA reported more variants than BWA + GATK: 250k–400k more per sample, and 229k more with respect to the union of all SNP sites of the three samples. More interestingly, the number of SNP conflicts detected by BALSA is 27k less than that of BWA + GATK; specifically, the conflict rate is 4.60% for BALSA and 5.47% for BWA + GATK. This shows that BALSA provides higher sensitivity and accuracy than BWA + GATK.

## Production testing on WGS—90 Chinese individuals

To test BALSA with the workload of a population scale study, we analyzed whole genome sequencing data of 45 CHB and 45 CHS samples from the 1000 genomes project (Table S2). In total, we have 90 samples of 100 bp paired-end reads with input size varying from 51.61 to 84.77-fold per sample (64.68-fold on average).

A computer cluster of 8 machines with three different hardware settings were used: (1) 5 machines with a 6-core Intel i7-3730k@3.2GHz + Nvidia GTX680, (2) 2 machines with a 6-core Intel E5-2620@2GHz + Nvidia GTX680, (3) 1 machine with a 6-core Intel E5-2620@2GHz + Nvidia Tesla K40. All 90 samples were analyzed by BALSA and with variants filtered by GATK VQSR. It took 3.13 days for the cluster to process all 90 samples (Table S3 shows the time consumption of each sample). Limited by the performance of the centralized storage for concurrent access by 8 machines, BALSA consumed more time on loading reads and writing results. From the statistics of run time on different hardware, we observed that a CPU with higher clock rate helps BALSA to better utilize the power of GPU. In order to utilize the extra power of newer GPU models, BALSA needs optimizations on the computation that utilizes CPU in the future.

The VCF files of the 90 individuals are available at http://www.bio8.cs.hku.hk/dataset/BALSA/90ChineseIdv/VCFs/.

## Somatic SNV and CNV detection on WGS—leukemia

We analyzed a pair of normal-tumor WGS sample on Donor Cell Leukemia. A previous study provides experimentally validated disease causing Somatic SNVs and CNVs on this paired sample (*Ho et al., 2012*). BALSA finished in 4.52 and 4.54 h for the normal (44.32-fold) and tumor (42.93-fold) sample, respectively. Using the SNAPSHOTs of the paired sample as input, BALSA's Somatic Mutation caller (Supplemental Information 9) finished in 16.47 min and generated 128,623 Somatic SNVs and 55,710 Somatic Indels passing the filter (Commands elaborated in Appendix 2.1.4). BALSA detected all the 16 Sanger validated disease causing SNVs (Table S4).

For comparison, we ran "SOAP3-dp + GATK", followed by two somatic mutation callers Mutect and SomaticSniper, which finished in 7.32 h and 1.14 h, respectively. When considering only functional changing mutations with types including "missense", "stop loss", "stop gain" and "splice site", BALSA, Mutect and SomaticSniper (*Larson et al., 2012*) identified 351, 2,945 and 8,963 somatic SNPs, respectively. Using the 16-genotype probabilistic model (Supplemental Information 8), which considers the coexistence of SNPs and Indels per site in a diploid space, BALSA effectively narrowed down the candidates of somatic variants for further investigation.

Table S5 shows the comparison between the experimentally validated somatic CNVs and the ones correspondingly detected by BALSA (Method in Supplemental Information 10). BALSA authentically detected the somatic CNVs with a fine-grain boundary in the validated regions (Table S6).

## WES—a 210x TCGA lung adenocarcinoma sample

We analyzed a 209.53-fold whole-exome sequenced TCGA lung adenocarcinoma sample (*The Cancer Genome Atlas Research Network, 2012*) (ID TCGA-44-7662) using BALSA. The pipeline finished in 24.65 min, identified 97,640 SNPs and 6,614 Indels passing the variant classification. Exome sequencing targets only tens of mega-bases of the genome; Where BALSA stores the SNAPSHOT file on a per-base basis for WGS, it stores only the user defined exome regions for WES in the purpose of storage saving (Supplemental Information 7).

## DISCUSSION

BALSA, as an extension of our GPU-based aligner SOAP3-dp, can finish the analysis of 50-fold whole genome sequencing data in a few hours; it was designed to favor the fast turn around time requirement for the clinical context. Unlike the traditional pipelines and tools that need to read and write Terabytes of intermediate data to the hard disk, BALSA performs the whole secondary analysis including quality control, alignment, base score recalibration, de-duplication and realignment in memory on the fly. With a neatly designed data structure, the analysis of a human genome costs only about 45 GB of memory, which makes BALSA applicable on most of the recent servers equipped with a commodity GPU.

BALSA was designed with sensitivity and accuracy prioritized over speed, and there still exists room for improving the speed of BALSA from an engineering perspective, such as

(1) design a more efficient pipeline to overlap the CPU tasks and GPU tasks; (2) reduce the data to be transmitted to and from GPU with better schema for reusing the data; (3) utilize new GPU features such as Hyper-Q to overlap multiple kernels to gain an even better hardware utilization. Better understandings on how the parameters affect the behaviors of the operating system also helps to improve the performance of BALSA in a long run (see Supplemental Information 2.5 for OS optimization guide).

Given BALSA's high efficiency, large genome centers may consider re-processing their historical sequencing data (say, thousands or even up to hundreds of thousands of samples) using BALSA so as to come up with standardized results for larger-scale genome analysis. Conventional thinking would suggest BALSA to store the alignment results of individual reads in BAM format or even the recently released CRAM (reference-based) format for later analysis. However, even if we just want to query a certain genome position over all the samples, the overhead in processing the alignment results in BAM or CRAM format is huge (the BAM format would demand a lot of time for decompression, and the CRAM format would require both decompression and recovering information from the reference). Suppose we have a hundred thousand samples, we estimate that using BAM or CRAM format, it would require several hours just to query a certain position of all the samples.

BALSA takes a different approach to store the alignment results for large-scale genome analysis. It stores a "SNAPSHOT" that records the per-base details with almost the same fidelity of a pileup from a BAM file. It allows much more efficient retrieval of per-base information, and it does not occupy much space, about 12 and 0.25 GB in size after LZ4 compression for a WGS and WES sample, respectively. BALSA's caller was designed to directly work on the "SNAPSHOT". Users can easily write their own downstream Apps utilizing SNAPSHOT, such as identifying SNPs and Indels from a SNAPSHOT or identifying somatic variants from multiple SNAPSHOTs (see Supplemental Information 7 for design and details), say, one may want to query the genotype frequency of 'GT' at a recurrent position in a tumor suppressor gene for those non-smoking female samples with age ranging from 50 to 80.

BALSA primarily focuses on the secondary analysis and takes input in the form of reads (FASTQ format). At present the process of preparing reads from a sequencer's raw signal, a.k.a. base-calling, relies almost exclusively upon vendor-provided software, such as Illumina's Bcl2FastQ (*Illumina*), which has been adopted by pipelines such as Mercury and ISAAC as a pre-processing before secondary analysis. Notice that, when compared with the time used by BALSA, the time consumed by such base calling software would become a bottleneck. To tackle the problem, some vendors are also using GPU to accelerate base calling; for example, in the Ion Proton platform (*Gupta & Siegel, 2013*).

BALSA can be easily integrated into existing workflows providing its simple interface. For better automation, BALSA will be improved to integrate external metadata resources and inputs such as a reference genome, sequence data locations, and a capture design bed file and therefore requires interaction with (Laboratory Information Management System) LIMS. To make BALSA portable, we will implement canonical APIs for transferring data

between BALSA and LIMS. These hooks are scripts that can be modified to query data from any metadata resource. LIMS and actively invoke BALSA when the sequencing data of a sample is ready. Examples of information served to BALSA from LIMS are the reference genome and gene regions.

The current implementation of BALSA assumes a computing node with a 6-core CPU, 40+ GB of memory and a GPU board. Such a configuration is pretty affordable to even small laboratories. Nevertheless, we have also considered how to make BALSA to run on other configurations, in particular, those available in public clouds like AWS. Very often cloud facilities may provide "computing instances", some of which with a lot of memory but no GPU, while others with too many GPUs but not enough memory. E.g., AWS provides a GPU instance "cg1.4xlarge", featuring 16 CPU cores, 22.5 GB memory, two Nvidia Tesla M2050 GPU devices, and 10 Gigabit inter-connectivity to other instances. To this end, we will implement an offload mode for BALSA so that BALSA can be run on two instances in parallel, one with sufficient memory but no GPU, plus one with two GPUs but insufficient memory. We expect that with suitable adjustment, the throughput of such implementation would be close to two copies of BALSA each running on a node with sufficient memory and a GPU.

### Funding
This work was funded by Hong Kong GRF (General Research Fund) HKU-713512E and ITF (Innovation and Technology Fund) GHP/011/12. The funders had no role in study design, data collection and analysis, decision to publish, or preparation of the manuscript.

### Grant Disclosures
The following grant information was disclosed by the authors:
Hong Kong GRF (General Research Fund): HKU-713512E.
ITF (Innovation and Technology Fund): GHP/011/12.

### Competing Interests
The authors declare there are no competing interests.

### Author Contributions

- Ruibang Luo conceived and designed the experiments, performed the experiments, analyzed the data, contributed reagents/materials/analysis tools, wrote the paper, prepared figures and/or tables, reviewed drafts of the paper.
- Yiu-Lun Wong performed the experiments, analyzed the data, contributed reagents/materials/analysis tools, prepared figures and/or tables, reviewed drafts of the paper.
- Wai-Chun Law performed the experiments, analyzed the data, contributed reagents/materials/analysis tools, reviewed drafts of the paper.

- Lap-Kei Lee and Jeanno Cheung analyzed the data, contributed reagents/materials/analysis tools, reviewed drafts of the paper.
- Chi-Man Liu analyzed the data, reviewed drafts of the paper.
- Tak-Wah Lam conceived and designed the experiments, wrote the paper, reviewed drafts of the paper.

## Supplemental Information

Supplemental information for this article can be found online at http://dx.doi.org/10.7717/peerj.421. The document is also available at http://www.bio8.cs.hku.hk/dataset/BALSA, along with the scripts used and results produced in the paper.

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
