# Peer review of "BALSA: integrated secondary analysis for whole-genome and whole-exome sequencing, accelerated by GPU"

_PeerJ, doi:10.7717/peerj.421_

## Round 0.1 · original submission · Minor Revisions

As you can see, both reviewers were very positive about your work and have requested only very minor changes. Based on their feedback, I would ask you to address the following points:

- One reviewer finds the Mendelian violation rate in the trio analysis higher than expected for both BALSA and GATK. Are there any reasons why this could be the case?

- Explain in the main text whether filtering was used in the trio study.

- Describe the size distribution of indels detected by BALSA and simulated indels in the main text.

- Move the summarized description of the workflow and the SNAPSHOT format to the main text from the supplementary material.

·

Basic reporting

The authors present a new whole genome and exome sequencing analysis pipeline that uses programming optimized for GPU processors to enable much faster analysis than most current methods. Importantly, they optimize all parts of the process to go from raw reads to variant calls, including homozygous reference calls. While it is much faster, one drawback is that it requires more memory than other methods.

Experimental design

It would be useful to describe what annotations the authors use in their random forest model in the main text of the paper.

It would be useful to describe the size distribution of indels detected by BALSA in the main text.

It would be useful to describe the size distribution of simulated indels in the main text.

For the trio study, it would be useful to say whether filtering was used in the main text. The Mendelian violation rate seems higher than I’d expect for both BALSA and GATK. I recommend that the authors manually investigate a subset of these errors to determine what might be causing them.

Since the NA12878 sample the authors analyzed is part of the Genome in a Bottle Consortium effort, the authors may find it useful to compare their calls to the high-confidence SNP, indel, and homozygous reference genotypes from this study (see the paper http://www.nature.com/nbt/journal/v32/n3/full/nbt.2835.html and most recent calls at http://genomeinabottle.org/blog-entry/new-high-confidence-na12878-genotypes-integrating-phased-pedigree-calls). This could help the authors estimate sensitivity and specificity in the high-confidence regions. It would also be useful for the authors to inspect manually the alignments around a subset of the discordant calls.

Validity of the findings

The conclusions are well-written, justified, and useful.

·

Basic reporting

The article is well written and meets all standards. Prior literature is appropriately referenced. I would prefer that the supplementary figures were included in the main text (unless there is a particular length restriction which prohibits this).
I would also like to see a summarized description of the workflow and the SNAPSHOT format in the main text, rather than the reader having to read this in the supplementary material.

Experimental design

The experimental design is adequate.

Validity of the findings

The data presented are robust and statistically sound.

Additional comments

This is an important piece of work, and is a genuine advance in the field.

---

## Round 0.2 · accepted · Accept

Thank you for addressing the reviewers' comments. I am now happy to accept your manuscript for publication at PeerJ.